# Real-Time Production Scheduling and Industrial Sonar and Their Application in Autonomous Mobile Robots

Francisco Burillo [1,*,†,‡], María-Pilar Lambán [1,‡], Jesús-Antonio Royo [1], Paula Morella [2] and Juan-Carlos Sánchez [2]

[1] Department of Design and Manufacturing Engineering, University of Zaragoza, 50018 Zaragoza, Spain; plamban@unizar.es (M.-P.L.); jroyo@unizar.es (J.-A.R.)

[2] TECNALIA, Basque Research Technology Alliance (BRTA), 50018 Zaragoza, Spain; paula.morella@tecnalia.com (P.M.); jcarlos.sanchez@tecnalia.com (J.-C.S.)

[*] Correspondence: 318190@unizar.es

[†] Current address: Departamento de Ingeniería de Diseño y Fabricación, Escuela de Ingeniería y Arquitectura, Universidad de Zaragoza, Campus Río Ebro, María de Luna, 3, 50018 Zaragoza, Spain.

[‡] These authors contributed equally to this work.

**Abstract:** In real-time production planning, there are exceptional events that can cause problems and deviations in the production schedule. These circumstances can be solved with real-time production planning, which is able to quickly reschedule the operations at each work centre. Mobile autonomous robots are a key element in this real-time planning and are a fundamental link between production centres. Work centres in Industry 4.0 environments can use current technology, i.e., a biomimetic strategy that emulates echolocation, with the aim of establishing bidirectional communication with other work centres through the application of agile algorithms. Taking advantage of these communication capabilities, the basic idea is to distribute the execution of the algorithm among different work centres that interact like a parasympathetic system that makes automatic movements to re-order the production schedule. The aim is to use algorithms with an optimal solution based on the simplicity of the task distribution, trying to avoid heuristic algorithms or heavy computations. This paper presents the following result: the development of an Industrial Sonar algorithm which allows real-time scheduling and obtains the optimal solution at all times. The objective of this is to reduce the makespan, reduce energy costs and carbon footprint, and reduce the waiting and transport times for autonomous mobile robots using the Internet of Things, cloud computing and machine learning technologies to emulate echolocation.

**Keywords:** real-time planning; production scheduling; Internet of Things; AMR; cyber-physical systems; smart manufacturing



## 1. Introduction

In real-time production scheduling, certain circumstances can arise that alter the ideal schedule that was previously established. These changes can occur at any time, and the ability to adapt to these changes is what determines success and survival. Sometimes these changes can be anticipated and preventative strategies can be put in place, but on other occasions, this is not possible due to external factors that cannot be controlled. In these cases, it is necessary to establish a working method capable of anticipating them. In animal life, it is the state of alert that allows us to face these circumstances immediately, and the response to them depends fundamentally on having considered the possible alternatives beforehand, in order to make the most effective decision as quickly as possible.

The technologies present in Industry 4.0 provide the necessary tools to maintain vigilance and respond quickly with the necessary required information. However, having more capabilities is helpful but not sufficient. The most important quality is to have the right strategy to extract the maximum performance from these capabilities. The aim of this

research work is to present an Industrial Sonar concept that provides the system with a reactive strategy to resolve any difficulties that may arise during production.

The article is structured as follows. Section 2 outlines the identification of the dynamic programming problem. Section 3 describes the materials and methods used in real-time production scheduling and discusses the decision-making process. Section 4 defines the main objectives and variables considered for the new strategy. Section 5 introduces the concept of the new Industrial Sonar method and its interaction with work centres, AMRs (Autonomous Mobile Robots) and AGVs (Automatic Guided Vehicles). Section 6 outlines the architecture of this new strategy. Section 7 presents the results obtained using this method in real production examples. Finally, Section 8 provides a brief conclusion and suggestions for future research work.

## 2. Working Hypothesis—Problem Identification

During day-to-day industrial processes, various problems may arise, such as machine malfunctions, human errors, damaged materials and external failures. These issues can cause delays in manufacturing operations and disrupt established planning, resulting in multiple problems at the business level, such as delayed delivery dates, increased costs and unmet demand.

Real-time scheduling is essential to deal with these types of issues. It allows problems to be dealt with promptly, either before, during, or immediately after they occur. Any delay in decision making results in economic cost and reduced quality and efficiency, which industries must try to limit in order to maximise benefits.

To deal with production problems that may arise during production, real-time scheduling can use alternative work centres. When doing so, it is important to consider the unique characteristics of each work centre, as they may have different capacities despite being alternatives. For example, in a scenario where a manufacturing lathe produces machine tools and has both expert and apprentice workers, there are likely to be significant differences in the resulting parts. These differences can be categorised as speed (process time), quality (features) and cost (labour and energy costs).

Production planning is usually conducted under ideal or standard conditions; that is, operations are planned assuming that both the expert resource and the apprentice will work under normal circumstances and will produce parts within a given time, cost and quality range, according to the expected average. However, in real time, multiple problems can arise, such as the expert having a bad day, being sick, having an accident, or encountering issues with the materials used. When these incidents arise, during real-time planning, the objectives being pursued must be clear at all times in order to act quickly. If the time, cost and quality variables are evaluated numerically, different decisions could have varying consequences for each of these variables. For instance, if the expert is unavailable, the apprentice could be assigned to complete a certain task, but this may result in lower-quality work taking longer to complete, although it could be less expensive. This decision has several implications, including the potential loss of customers, cost savings, potential financial losses, if repairs are necessary, and delays.

This problem becomes significantly more complex when there are multiple work resources, materials, machinery and tools, as well as different AMR models of varying capacities. Additionally, external factors can also impact more complex production processes. To enable real-time planning in these more complex production environments, current technologies provide tools for numerical evaluation of these variables in real time and the necessary changes and adaptations in these environments. To achieve this, Industry 4.0 technologies are utilised, including the digital transformation of production through cyber-physical systems. These industrial control systems use software capable of utilising the potential of different technologies, such as the IoT (Internet of Things), RDFI (radiofrequency), robotics, and machine learning, to achieve greater automation, convergence of the physical and digital worlds, delegation of certain decisions to machines, data retrieval, and increased product customisation.

Industry 4.0 [1] is a convergence of existing and emerging technologies that interact with each other, creating a dynamic and constantly evolving concept. Therefore, Industry 4.0 encompasses various technologies including cloud computing, big data, artificial intelligence, data analysis, edge computing, mobiles, data network technologies, ICSs (industrial control systems) (HMI—human–machine interface; SCADA—industrial automation), business control systems (MESs—manufacture execution systems; ERP—enterprise resource planning), sensors and actuators, MEMSs (microelectomechanycal systems) and transducers and AMRs. The algorithm design for this research work aims to be open, allowing for real-time collaboration between different technologies.

Real-time planning needs to answer several key questions. First, what decisions need to be made? Second, based on the selected variables, how should these decisions be made using the appropriate technology? Finally, when should these decisions be made? Answering these questions is critical in real-time planning, where in-process operations may need to be rescheduled if certain circumstances arise during production.

In a production order, the variables of time, cost and quality are interrelated, making it complex to act in real time on one of them without affecting the others. Therefore, it is important to carefully consider what, how and when to make a decision. When planning in real time, it is important to select the appropriate variables and analysis method, apply the selected technology and make the best decisions in the shortest time possible.

## 3. Materials and Methods

"What" decision should be made in real-time production scheduling? Traditional planning systems are not effective at providing a solution to the continuous variations that occur in real-time production scheduling. As a result, various dynamic planning strategies have emerged that use new systems and technologies to propose the best possible solution based on the data obtained.

Each strategy focuses on specific variables, objectives and methods. These variables are obtained through various technologies previously referred to as sensors, business systems and big data. For instance, sensors can capture production data to calculate production costs and evaluate production quality in real time.

The fundamental strategy for decision making during planning defines the objectives. Some strategies prioritise the objective of energy cost reductions, while others prioritise process time reductions. Some strategies are multi-objective.

"How" to make real-time scheduling decisions? To determine how to schedule, certain methods are defined and applied, typically based on heuristic and metaheuristic algorithms. These methods aim to find the best possible solution for the given variables, problems and objectives.

Production planning involves planning and sequencing the various operations of production orders across different work centres. When manufacturing a product, all operations may take place in a single work centre or involve several work centres.

There are methods available for considering the operations of a single work centre in production scheduling. Some of the scheduling methods that can be applied with the previous methods and variables are the SPT (shortest process time), the R&M algorithm (Rachamadugu and Morton), Hogdson, and method weighted methods [2].

At the same time, there are methods that consider the operations of multiple work centres in production planning. These methods take into account the impact of other production operations that must be completed in other work centres. They evaluate different combinations based on the number of orders, quantity, time and characteristics. Depending on the complexity of the combinations, some planning problems can be solved polynomially, and others cannot be solved and are classified as NP, NP-complete or NP-hard problems. In the NP cases, the search can be completed with the optimal solution, while in other cases, the search can be completed with a heuristic solution or a good possible solution [3–5]. The main methods or strategies used by the authors can be classified as shown in Table 1.

**Table 1.** Dynamic planning method classification.

| Strategy | Method |
|---|---|
| Multi-period hierarchical mechanism | Luo et al. [6] |
| Multi-agent architecture | Shukla et al. [7] |
| Genetic algorithm | Fang et al. [8] |
| Games theory | Jin Wang et al. [9] |
| Swarm intelligence approach | Rossi [10] |
| Gravitational emulation local search algorithm | Hosseinabadi et al. [11] |
| ICM—Intelligent Collaborative Mechanism and Optimization Algorithm | Quian et al. [12] |
| Planning algorithm based on Petri net and ant colony heuristic algorithm | Tian et al. [13] |
| Software designed in network | Jha et al. [14] |
| Algorithm in cloud ERP | Navin et al. [15] |
| Neighbourhood heuristic (IVNH) algorithm | Otto et al. [16] |
| Algorithm based on an improvement of particle swarm optimization | Tang et al. [17] |
| MDSS algorithm (uncertainty) | Jiang et al. [18] |
| Heterarchical architectures | Duffie et al. [19] |
| Zero-defect manufacturing (ZDM) | Psarommatis et al. [20] |
| Augmented reality application and holonic approach to adapt production rescheduling | Mourtzis et al. [21,22] |

- Multi-period hierarchical mechanism: Luo et al. [6] proposed a multi-period hierarchical mechanism to optimise production planning and real-time planning for a hybrid workshop flow.
  The multi-period hierarchical scheduling (MPHS) mechanism was developed based on the decision procedure, in which the proposed scheduling mechanism contains two levels (shop floor decision and work stage decision) in order to control the workload balance.
- Multi-agent architecture: Shukla et al. [7] developed a multi-agent architecture for a workshop flow.
  An agent-based architecture was developed that enables the integration of production planning and scheduling, facilitating real-time production scheduling to implement the concept of MAS.
- Genetic algorithms (G. Chryssolouris, V. Subramaniam; *Journal of Intelligent Algorithms*) [23]: the grid method and genetic algorithm (Q. Yang, M. Yu, S. Liu) [24], the genetic algorithm with particle swarm optimisation (Wang et al.) [25], and the multi-objective optimisation algorithm (Fang et al.) [8].
  A genetic algorithm (GA) is a method commonly used in the real-time scheduling solutions. In this type of metaheuristic algorithm, the optimisation of search problems is inspired by the process of natural selection, which mimics biological evolution. It belongs to the group of evolutionary algorithms.
- Game theory: Jin Wang et al. [9] proposed an algorithm based on game theory that considers energy consumption.
  The application of the game theory was based on the idea of using mathematical models of strategic interaction between rational decision makers. Mathematical models of real-time scheduling can be established to improve the production efficiency and reduce energy consumption, based on an infinitely repeated game-stage model.

- Swarm intelligence approach (Rossi) [10].
  Swarm intelligence (SI) algorithms are based on the collective behavior of decentralised, self-organised systems. They are based on natural systems, but can be applied to artificial systems. They consist of a population of simple agents interacting locally with one another and with their environment.
- Gravitational emulation local search algorithm (Hosseinabadi et al.) [11].
  To obtains multi-objective solutions to job shop scheduling problems, heuristic algorithms based on the gravitational emulation local search algorithm (GELS) can be used to effectively search the problem space and generate near optimal and quality solutions for seeking and solving NP-complete problems.
- Intelligent Collaborative Mechanism (ICM) and optimisation algorithm, Quian et al. [12].
  In order to reduce the impact of the exceptions to the dynamic scheduling, the Intelligent Collaborative Mechanism can be used where negotiations on resource configuration can take place between tasks based on a data-driven ICM framework, Petri-net-based analysis and constraint matrix.
- Planning algorithm based on the Petri net and ant colony heuristic algorithm, Tian et al. [13]. The ant colony algorithm is a probabilistic technique that can be used to solve computational problems that can be reduced to finding the best paths through graphs. Petri net can be used to manufacture graphs. This constitutes metaheuristic optimisation in swarm intelligence methods.
- Software designed in network, Jha et al. [14]. Specific software can be designed to model the Industrial Internet of Things (IIoT) for manufacturing-based applications, analysing resource failures and repair capabilities, transferring work in progress to other processing resources at the same or a remote location, and replacing resources by purchasing new ones.
- Algorithm in cloud ERP, Navin et al. [15]. ERP programs can be used to develop specific algorithms in order to work with multiple variables and multiple objectives in real-time dynamic production planning. The ERP enables the possibility of including data not strictly related to the production area in real-time algorithms, and covers many more variables and objectives indirectly related to production.
- Neighbourhood heuristic (IVNH) algorithm, Otto et al. [16]. This is another meta-heuristic method can be used to solve a range of combinatorial optimisation problems for manufacturing. It explores distant neighbourhoods of the current solution to solve linear and nonlinear problems.
- Algorithm based on an improvement to particle swarm optimisation, Tang et al. [17]. Based on particle swarm optimisation (PSO), this is a computational method that optimises a problem by iteratively trying to improve a candidate solution with respect to a given quality measure. Some improvements can be adapted to apply this metaheuristic algorithm to manufacturing processes for dynamic production planning purposes.
- MDSS algorithm to solve manufacturing cases in planning problems where the process and arrival times are intervals (uncertainty), Jiang et al. [18]. In cases of uncertain scheduling problems where the process and arrival times are in intervals, a multi-stage and knowledge-based algorithm can be considered.
- Heterarchical manufacturing systems, (Duffie) [19], contract net protocol (Smith) [26], real-time heterarchical control of dynamic resource allocation and dynamic product routing using a NetLogo simulation Zbib [27].
  Heterarchical architectures are capable of modelling systems based on the interactions between the entities in manufacturing systems. This model and its application are proposed in the real-time heterarchical control of dynamic resource allocation and dynamic product routing.
- Zero-defect manufacturing (ZDM) methodology: (Psarommatis) [20].
  Based on traditional and ZDM-oriented events, a methodology was proposed to deal with rescheduling caused by unexpected events. This methodology uses parameters

to calculate when the next rescheduling will be performed to effectively respond to these events.

- Augmented reality application and a holonic approach to adapting production rescheduling: (Mourtzis) [21,22].

  Based on mobile device interfaces and using augmented reality, a holonic approach was proposed, aiming to support dynamic production rescheduling by representing the construction of every holonic manufacturing system in a bottom-up manner by integrating holons that collaborate to provide flexibility and robustness.

The strategy followed by the planner in the face of real-time production planning problems involves deciding on the combination of the most appropriate technologies and techniques that can cover the objectives persuaded.

To these techniques, we can add an example of a scheduling algorithm with an optimal solution: Johnson's method. This method can be used to schedule n jobs on two machines, with the objective of minimising the time required to complete the n jobs in the workshop (makespan), which can be solved optimally.

When scheduling jobs on more than two machines, the problem becomes increasingly complex and requires more advanced algorithms. Some examples of effective planning algorithms, each with their own best heuristic or meta-heuristic solution, include simulated annealing algorithms, genetic algorithms, evolutionary algorithms, the particle swarm optimisation algorithm, and the artificial bee colony algorithm. In summary, on the basis of the variables introduced, the objectives pursued and the methods and algorithms used, a new plan for the sequence of operations to be carried out in each centre is proposed for all the work centres involved.

When considering "how to make real-time scheduling decisions", it is important to explore the communication possibilities offered by Industry 4.0 technologies. Once the variables to be analysed have been defined, sensors and actuators can be used to obtain the necessary values during production. For example, real-time values of process variables, total manufacturing costs (including material costs and energy costs) and certain variables for assessing product quality can be captured.

An individual analysis of variable values can be used to apply real-time planning methods to achieve the objectives. For example, cyber-physical systems can be combined with additional software that applies specific algorithms. These values can also be incorporated into business systems such as MESs and ERP, where they can be complemented with a greater number of data variables outside of production. This enables real-time planning by applying strategies with a wider range of objectives.

"When" to make decisions in real-time scheduling?

Real-time scheduling is defined as dynamic planning by definition. It is the result of a continuous interaction between real-time variable data and the application of methods and algorithms to achieve the desired objectives. Therefore, a continuous analysis of the results and a continuous reevaluation of the most appropriate plan to maintain these objectives are necessary.

AMRs and AGVs are used for material handling in the manufacturing process and are responsible for connecting production processes. The control of these robots is essential for real-time production scheduling to connect the tasks in different work centres. Therefore, finding the optimal transport time and equipment for each task with respect to the production specifications is crucial in production planning. AGVs operate with minimal integrated intelligence and move in a guided manner through a closed circuit. In contrast, AMRs rely on software that provides them with real-time maps and do not require a circuit to guide their movement. Optimising the transportation time and allocating both AGVs and AMRs to production logistics tasks are important objectives to be incorporated into real-time programming [28]. Therefore, it is important to consider the most commonly used rules for dispatching AGVs in the real-time scheduling problem, as suggested by Klei and Kim [29].

With this consideration, transport robots as AGVs have been incorporated into the application of heuristic algorithms using different strategies.

Ulusoy et al. [30] proposed a genetic algorithm approach to solve the problem of simultaneous scheduling of machines, and proposed using a number of identical AGVs in an FMS to minimise the makespan. Zhang et al. [31] proposed an allocation strategy and dynamic optimisation method for shop floor material handling based on multi-source manufacturing data. Saidi-Mehrabad et al. [32] proposed a mixed-integer linear programming approach and applied a two-stage ant colony algorithm (ACA) to solve a model of job shop scheduling problems and a conflict-free routing problem for AGVs. Mousavi et al. [33] proposed a multi-objective scheduling approach for AGVs in a flexible manufacturing system (FMS) using a hybrid of the genetic algorithm and the particle swarm algorithm. This approach considers multiple objectives, including minimising the makespan and the number of AGVs. Hu et al. [34] proposed a real-time scheduling approach using deep reinforcement learning to reduce the makespan and delay ratio of AGVs. Liu et al. [35] proposed a multi-population co-evolutionary algorithm to solve a green multi-objective IPPS problem considering a logistics system. This algorithm aims to minimise the energy consumption and maximise the completion time.

## 4. Objectives and Variables for New Strategies

When analysing the state of the art in real-time production scheduling, it can be observed that a wide range of objectives and variables are used to solve the possible problems that may arise in dynamic planning.

An analysis of the literature reveals that the main analysis variables are process time, production cost, energy consumption, priority, delivery date, delivery time and transport.

Based on these variables, objectives have been set, including minimising the makespan (Cmax), reducing the machine load, reducing the energy consumption and carbon footprint, minimising the total cost, reducing delivery date delays, minimising the delivery time in manufacturing and reducing the transport time.

This research work proposes a new strategy for the application of dynamic algorithms that provide a real-time planning solution. The solution typically involves the application of heuristic algorithms to obtain the best possible solution.

This work aims to solve the problem of real-time planning from a new perspective, using the principle of 'divide and conquer' (also known as 'divide et impera', a famous phrase attributed to Julius Caesar). Instead of using a single algorithm to process all the available information, it has been decided to utilise current communication technology to apply simple algorithms that can work together to provide an optimal solution, rather than an approximation. These algorithms will be applied in different work centres through communication between them.

As discussed earlier, the number of variables that can be used and exploited depends on the methods used and the availability of data. Therefore, authors who use specific software or ERP as the basis for their algorithms consider a larger number of variables than others.

## 5. Industrial Sonar Algorithm

The previous section covered the main objectives and variables that are used for optimal dynamic planning. This research proposes a new strategy that focuses on providing algorithms and solutions for dynamic planning.

Dynamic production scheduling involves the application of different algorithms to operations that need to be performed at different work centres in an industrial environment. The aim is to optimise the production sequence by minimising several aspects, including the makespan (the time between the start and finish of all jobs), the process time of each task, the number of tasks in process, the delay time, the cost of unsatisfied demand, environmental costs and transport times.

The production schedule in each work centre can be altered by multiple circumstances that occur on a daily basis, such as machine stops, defective components and quality problems. These circumstances can significantly affect the desired production results.

Therefore, it is essential to plan real-time production processes at the different work centres taking these circumstances into account. Current technologies enable communication and interaction in the workplace.

Any delay or stoppage in the plan of a work centre is considered an obstacle. For example, certain animals have developed methods to overcome physical obstacles in nature. Dolphins and bats, for instance, have evolved an echolocation system. They emit sounds and interpret the echoes that bounce back from the obstacles they encounter. This enables them to detect objects and obstacles around them and make decisions about where to go. The sonar system of ships and submarines is based on the same principle. By using this principle as a strategy, we can establish communication with the next work centres in the production sequence to gather information about their shipments or shipments to alternative work centres. This can be achieved by treating it as an echolocation or Industrial Sonar system.

This process is a reactive strategy for real-time planning of production. It can be applied at the end of each production operation. Each work centre can also be proactive in terms of placing current production orders in a queue at other work centres. The same strategy can be applied to any problem that may arise during production, such as a machine stoppage.

Prior to implementing this reactive strategy, an initial daily production planning phase may be required. During this phase, a specific algorithm will sequence the production schedule at each work centre. However, this step is not strictly necessary.

## 6. Industrial Sonar Architecture

The objective is to achieve real-time planning, allowing for the resolution of any issues that may arise during production. This involves re-planning tasks and tasks at other work centres using algorithms that have been designed previously, in addition to automatic communication between work centres.

The methodology used is the application of the Industrial Sonar concept (Figure 1). This is based on establishing communication between machines to request the execution of a simple algorithm, allowing for quick execution. Each machine completes a task while simultaneously carrying out a series of actions or communications with other work centres:

- Communication 1: An information request is sent to the next work centre and its alternative work centres, where the next operation of the production order after the completed operation must be carried out. For example, if operation 10 has been completed, a request is made to the different work centres in which it is possible to carry out operation 20. This request for information requires the application of a specific algorithm to order the tasks of the destination work centre, including the task that can be submitted. A response is requested for the sequence of the operations, applying the algorithm indicated and the information regarding the date and time at which the operation can be completed. The scheduling algorithm used may be the SPT (shortest processing time) or any other work centre scheduling algorithm such as FE, CEF, or ECF, which is an algorithm designed to reduce energy costs.
- Communication 2: On the other hand, a request for information is sent to the alternative work centres where the operation following the current one can be carried out. For example, if the next operation is 20, the request is sent to the work centres where operation 30 can be carried out. This request for information requires the application of a specific algorithm to order the operation tasks at the destination work centre, including the task that is to be submitted. A response is requested for the sequence of the operations, applying the algorithm indicated and the information regarding the date and time at which the operation can be completed. The algorithm may be the SPT (shortest processing time) or any work centre scheduling algorithm such as FE, CEF, or ECF, which is an algorithm designed to reduce energy costs.
- Application of the Industrial Sonar algorithm: Specific algorithms designed for job shop programming can be applied based on the responses of the work centres.

To achieve the proposed objectives, the responses of the work centres operating at 20 and 30 can be combined in a specific way.

For example, the objective of achieving the lowest makespan, i.e., the shortest total production time of all the operations involved, can be achieved using the Johnson algorithm. This involves initially sequencing the SPT alternately in the work centres for operation 20+ and, at the end, in the work centres of operation 30. This approach allows us to make an optimal decision that minimises the makespan.

For more complex examples where the work centre algorithms are invoked, the work centre operations can be organised based on a combination of the makespan, the lowest energy consumption and the lowest carbon footprint. This is similar to using Johnson's algorithm to optimise the makespan and energy losses, and also incorporates additional criteria to achieve the desired objectives. This may include minimising yjr transport time by communicating with the AMRs responsible for managing transport between work centres.

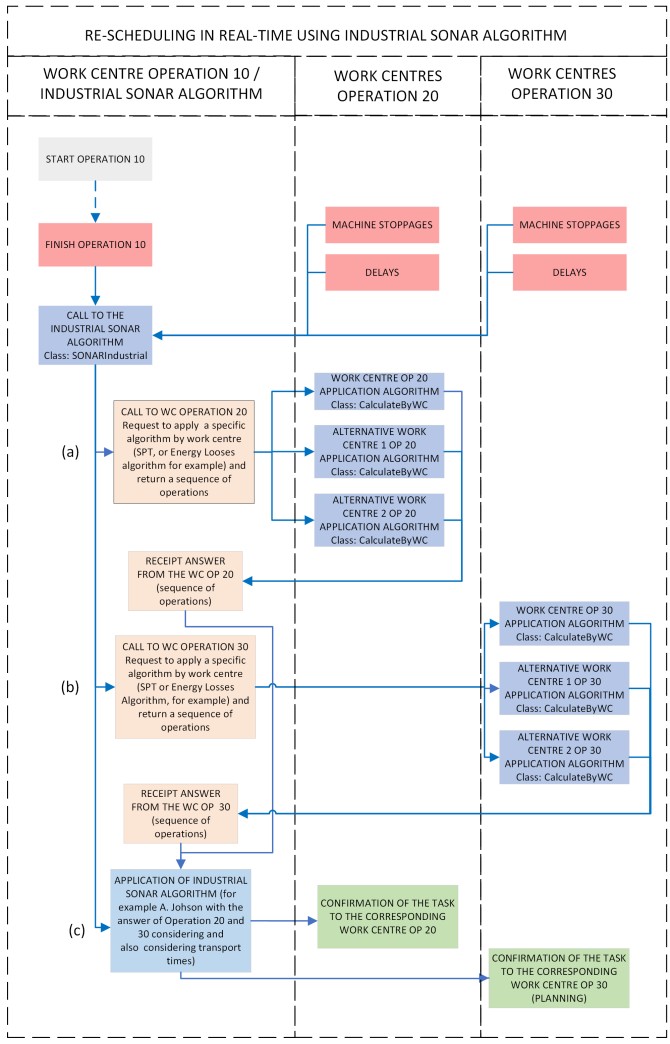

**Figure 1.** Communication process (1 and 2) -> Industrial Sonar algorithm -> decision making. This figure explains the communication process of the Industrial Sonar algorithm: (**a**) Firstly, Communication 1 is established with work centres of operation 20 to obtain an answer regarding the schedule of the tasks. (**b**) Simultaneously, Communication 2 is established with work centres of operation 30 to obtain an answer also regarding the schedule of the tasks. (**c**) With the operation schedules of each work centre obtained, the Industrial Sonar algorithm is applied and a decision is made, selecting and confirming one specific work centre.

The eco-sequencing algorithm is launched by each work centre as soon as any capacity saturation caused by production problems is detected. This is similar to the proposed use of the Industrial Sonar algorithm (ISA) at the end of the production order operation in the work centre.

To measure the time, machine failures and quality problems, as well as manage algorithm application and responses, the following hardware and software architecture has been designed. This enables real-time execution of planning algorithms at each work centre. The algorithms used are technology-independent and can be adapted to various system architecture designs.

- Algorithm execution device: To execute the algorithms, a computer or IoT device with sufficient power is required. During testing, the Raspberry Pi 4 (Raspberry Pi Foundation, 2022) and Jetson Nano (NVIDIA, 2022) were selected due to their size and availability.
- Sublime Text 3: This was used as the software editor for the development of the Python programme (Sublime Text, 2022).
- Python version: Python 3.10.5. (Python Software Foundation., 2022).
- SQLite: The software uses the SQLite database (SQLite, 2022) for its versatility in keeping all the necessary information on the device concerning the operations within the work centre.

## 7. Results and Discussion

The analysis of the results obtained using the Industrial Sonar method was carried out on a real manufacturing process, consisting of a sequence of operations performed at two different work centres: a Haas numerical control (CNC) machine and a lathe. Each of these machines has an alternative work centre. Additionally, the lathe has both an expert worker and an apprentice.

A Python-based method was developed and used to schedule production tasks at the work centre. This method sequences the production operation tasks of a work centre based on historical data on the previous production losses of the item in the work centre. From these historical data, the system captures energy losses based on pieces, energy losses based on OEE (overall equipment effectiveness) and the carbon footprint. These data are considered in the algorithm's calculation (Figures 2 and 3).

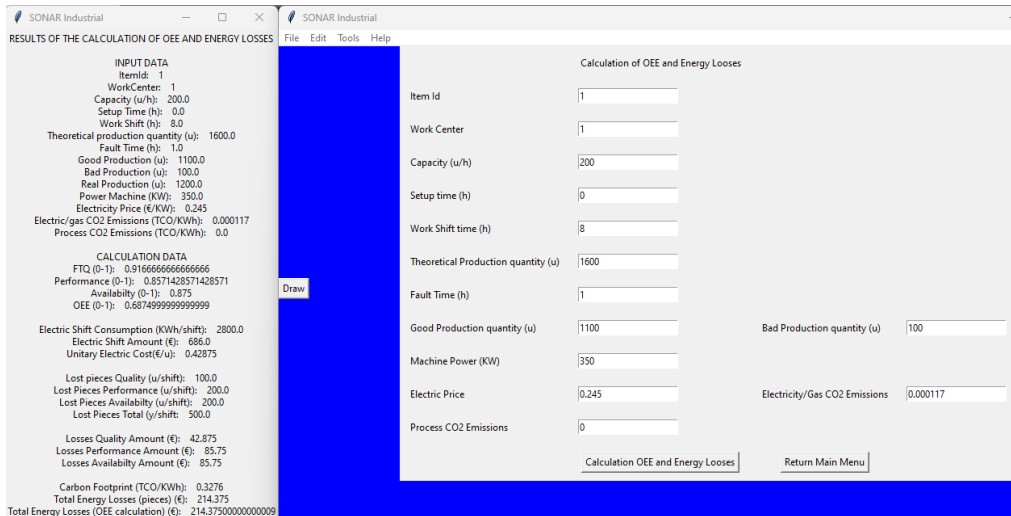

**Figure 2.** Energy losses calculated for the item and work centre. This figure illustrates an example of the energy losses calculated for one item and work centre, with all the different terms of the calculation detailed: the energy losses based on pieces, energy losses based on OEE (overall equipment effectiveness) and the carbon footprint.

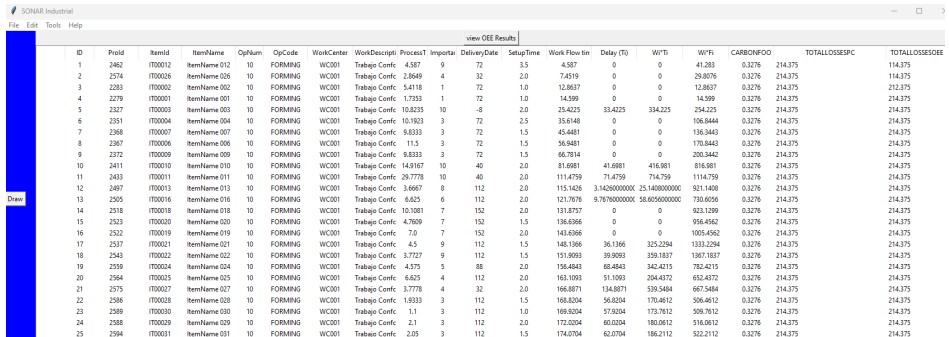

**Figure 3.** Historical data on energy loss calculations for the items in the work centre. This figure presents an example of historical data registered on energy loss calculations for the items in the work centre. These historical data are the input for an algorithm to schedule the production operations by energy losses.

A communication strategy was designed and developed in Python to facilitate the exchange of requests and responses between work centres. The strategy involves sending production order data to alternative work centres for the following operation, with a request to apply a specific algorithm based on the combination of the SPT and energy losses based on OEE criteria (Figure 4).

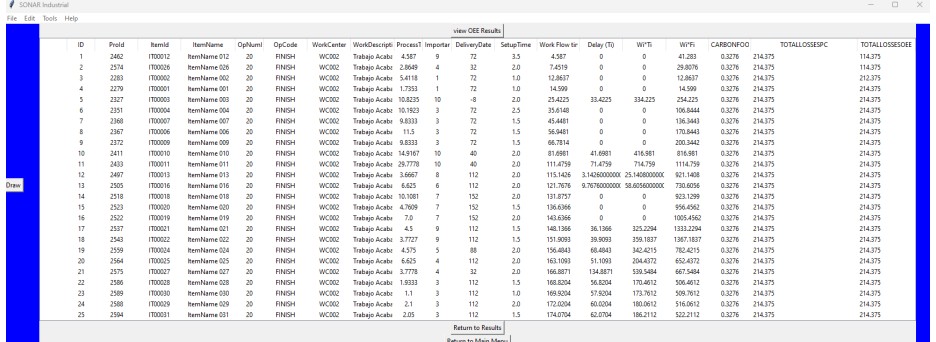

**Figure 4.** Sequence of production tasks ordered according to requested criteria of energy losses. This figure displays an example of the sequence of production tasks returned by a work centre for the next operation, ordered according to the requested criteria of SPT + energy losses based on OEE). These data are the result of Communication 1.

Simultaneously, the production data are communicated to the work centres of the following operation (to the next operation) requesting the application of a specific algorithm that combines the SPT and energy losses based on OEE criteria (Figure 5).

**Figure 5.** Production tasks ordered based on criteria (OEE + transportation time). This figure displays an example of the sequence of production tasks returned by a work centre for the operation following the current operation (the next operation) with a specific order based on the requested criteria (SPT-OEE + transportation time). These data are the result of Communication 2.

As an example of real-time planning, the Industrial Sonar algorithm was applied to the previous data. This algorithm was designed to reduce the makespan and the energy costs. A calculation was included in the algorithm to reduce the transportation time between AMRs. The following results were obtained (Figure 6).

| ID | ProId | ItemId | ItemN | OpNu | OpCode | WorkCenter | WorkDescript | Process | Importa | DeliveryDate | SetupTime | Work Flow tir | Delay (Ti) | Wi*Ti | Wi*Fi | CARBONFOO | TOTALLOSS | TOTA | CMAX |
|---|---|---|---|---|---|---|---|---|---|---|---|---|---|---|---|---|---|---|---|
| 124 | 2575 | IT00027 | ItemN | 20 | FINISH | WC002 | Trabajo Acab | 3.7778 | 4 | 32 | 2.0 | 33.6516 | 1.6516 | 6.6064000000 | 134.6064 | None | None | None | 291.4! |
| 125 | 2510 | IT00017 | ItemN | 20 | FINISH | WC002 | Trabajo Acab | 4.3214 | 6 | 112 | 2.0 | 37.973 | 0.0 | 0.0 | 227.838 | None | None | None | 295.7 |
| 126 | 2496 | IT00014 | ItemN | 20 | FINISH | WC002 | Trabajo Acab | 4.5 | 8 | 112 | 2.0 | 42.473 | 0.0 | 0.0 | 339.784 | None | None | None | 300.2 |
| 127 | 2537 | IT00021 | ItemN | 20 | FINISH | WC002 | Trabajo Acab | 4.5 | 9 | 112 | 1.5 | 46.973 | 0.0 | 0.0 | 422.757 | None | None | None | 304.7 |
| 128 | 2559 | IT00024 | ItemN | 20 | FINISH | WC002 | Trabajo Acab | 4.575 | 5 | 88 | 2.0 | 51.548 | 0.0 | 0.0 | 257.74 | None | None | None | 309.3 |
| 129 | 2462 | IT00012 | ItemN | 20 | FINISH | WC002 | Trabajo Acab | 4.587 | 9 | 72 | 3.5 | 56.135 | 0.0 | 0.0 | 505.215 | None | None | None | 313.9 |
| 130 | 2600 | IT00091 | Item I | 20 | FINISH | WC002 | Trabajo Acab | 4.741 | 1 | 40 | 1.5 | 60.876 | 20.876 | 20.876 | 60.876 | None | None | None | 318.6 |
| 131 | 2523 | IT00020 | ItemN | 20 | FINISH | WC002 | Trabajo Acab | 4.7609 | 7 | 152 | 1.5 | 65.6369 | 0.0 | 0.0 | 459.4583 | None | None | None | 323.4 |
| 132 | 2283 | IT00002 | ItemN | 20 | FINISH | WC002 | Trabajo Acab | 5.4118 | 1 | 72 | 1.0 | 71.0487 | 0.0 | 0.0 | 71.0487 | None | None | None | 328.8 |
| 133 | 2505 | IT00016 | ItemN | 20 | FINISH | WC002 | Trabajo Acab | 6.625 | 6 | 112 | 2.0 | 77.6737 | 0.0 | 0.0 | 466.0422 | None | None | None | 335.4 |
| 134 | 2564 | IT00025 | ItemN | 20 | FINISH | WC002 | Trabajo Acab | 6.625 | 4 | 112 | 2.0 | 84.2987 | 0.0 | 0.0 | 337.1948 | None | None | None | 342.0 |
| 135 | 2522 | IT00019 | ItemN | 20 | FINISH | WC002 | Trabajo Acab | 7.0 | 7 | 152 | 2.0 | 91.2987 | 0.0 | 0.0 | 639.0909 | None | None | None | 349.0 |
| 136 | 2607 | IT00035 | ItemN | 20 | FINISH | WC002 | Trabajo Acab | 7.1786 | 10 | 32 | 2.5 | 98.4773 | 66.4773 | 664.773 | 984.773 | None | None | None | 356.2 |
| 137 | 2501 | IT00015 | ItemN | 20 | FINISH | WC002 | Trabajo Acab | 8.7188 | 6 | 112 | 2.0 | 107.1961 | 0.0 | 0.0 | 643.1766 | None | None | None | 364.9 |
| 138 | 2368 | IT00007 | ItemN | 20 | FINISH | WC002 | Trabajo Acab | 9.8333 | 3 | 72 | 1.5 | 117.0294 | 45.0294 | 135.0882 | 351.0882 | None | None | None | 374.8 |
| 139 | 2372 | IT00009 | ItemN | 20 | FINISH | WC002 | Trabajo Acab | 9.8333 | 3 | 72 | 1.5 | 126.8627 | 54.8627 | 164.5881 | 380.5881 | None | None | None | 384.6 |
| 140 | 2518 | IT00018 | ItemN | 20 | FINISH | WC002 | Trabajo Acab | 10.1081 | 7 | 152 | 2.0 | 136.9708 | 0.0 | 0.0 | 958.7956 | None | None | None | 394.7 |
| 141 | 2351 | IT00004 | ItemN | 20 | FINISH | WC002 | Trabajo Acab | 10.1923 | 3 | 72 | 2.5 | 147.1631 | 75.1631 | 225.4893 | 441.4893 | None | None | None | 404.9 |
| 142 | 2327 | IT00003 | ItemN | 20 | FINISH | WC002 | Trabajo Acab | 10.8235 | 10 | -8 | 2.0 | 157.9866 | 165.9866 | 1659.866 | 1579.866 | None | None | None | 415.7 |
| 143 | 2367 | IT00006 | ItemN | 20 | FINISH | WC002 | Trabajo Acab | 11.5 | 3 | 72 | 1.5 | 169.4866 | 97.4866 | 292.4598 | 508.4598 | None | None | None | 427.2 |
| 144 | 2371 | IT00008 | ItemN | 20 | FINISH | WC002 | Trabajo Acab | 11.5 | 3 | 72 | 1.5 | 180.9866 | 108.9866 | 326.9598 | 542.9598 | None | None | None | 438.7 |
| 145 | 2411 | IT00010 | ItemN | 20 | FINISH | WC002 | Trabajo Acab | 14.9167 | 10 | 40 | 2.0 | 195.9033 | 155.9033 | 1559.033 | 1959.033 | None | None | None | 453.7 |
| 146 | 2626 | IT00036 | ItemN | 20 | FINISH | WC002 | Trabajo Acab | 15.3333 | 1 | 256 | 2.0 | 211.2366 | 0.0 | 0.0 | 211.2366 | None | None | None | 469.0 |
| 147 | 2363 | IT00005 | ItemN | 20 | FINISH | WC002 | Trabajo Acab | 16.7857 | 3 | 72 | 2.5 | 228.0223 | 156.0223 | 468.0669 | 684.0669 | None | None | None | 485.8 |

**Figure 6.** Outcome of the Industrial Sonar algorithm using 'Sonar Industrial' software developed in Python. This figure displays the outcome of applying the Industrial Sonar algorithm using 'Sonar Industrial' software developed in Python to test the hypothesis of this research. This is a result of applying the Industrial Sonar algorithm with the data returned from the communications with the work centres in Figures 4 and 5.

The following graphs illustrate the relationship between the Cmax and energy losses objectives (Figure 7). The optimal solution is compared under three different conditions: Firstly, by considering only the objective of Cmax (minimising makespan). Secondly, by considering only the order of energy losses. Finally, by considering both objectives, minimizing Cmax and ordering by energy Losses, through the application of the Industrial Sonar algorithm.

The optimal solution considers only the Cmax objective using Johnson's algorithm.

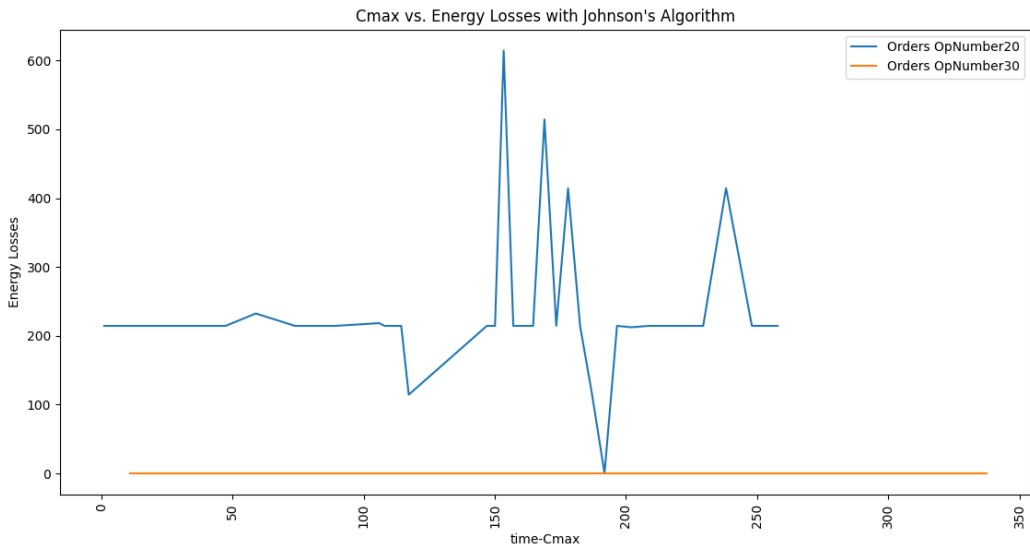

**Figure 7.** Graph of the application of the Johnson's algorithm to optimise Cmax. This figure shows the relationship between Cmax and energy loss objectives in the work centre for all the operations scheduled applying Johnson's algorithm. In this case, in the application of Johnson's algorithm, only optimisation of Cmax values is performed.

The optimal solution considering only the objective of energy losses, without applying Johnson's algorithm, is shown in Figure 8.

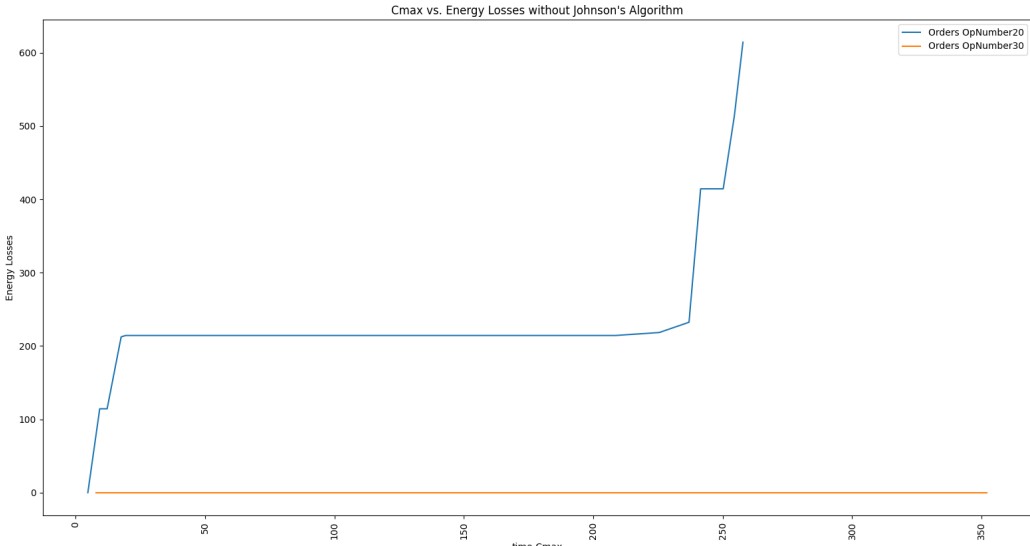

**Figure 8.** Graph of the application of the algorithm to reduce and optimise energy losses. This figure shows the relationship between Cmax and energy loss objectives in the work centre for all the operations scheduled, applying an algorithm to reduce and optimise energy losses.

The proposed optimal solution considering both the objective of reducing Cmax and energy losses using the Industrial Sonar algorithm is shown in Figure 9.

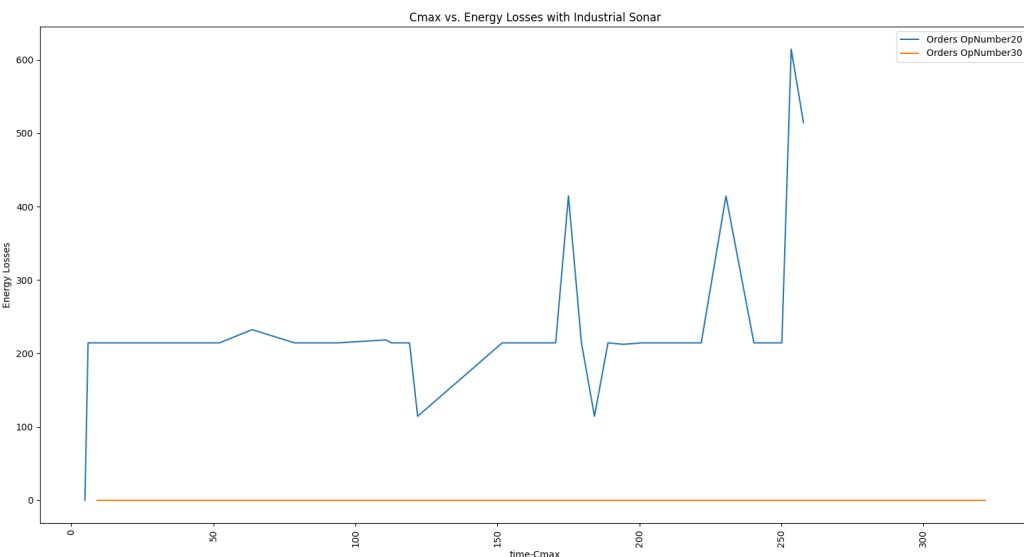

**Figure 9.** Graph of the application of the Industrial Sonar algorithm to reduce and optimise Cmax and energy losses. This figure shows the relationship between Cmax and energy loss objectives in the work centre for all the operations scheduled, applying an Industrial Sonar algorithm to reduce and optimise the Cmax, energy loss and transport time.

A comparative analysis was conducted on Figures 7–9 to contrast their errors with respect to the optimal solution of each variable. It was observed that Figure 9, which applies the Industrial Sonar algorithm, uses a slightly higher Cmax to complete production operations than Figure 7, in which Johnson's algorithm is applied. However, in exchange for this, a more uniform distribution of operations can be achieved to optimise the energy loss. To achieve this, it is recommended to distribute operations to slots with lower energy

costs. Comparing Figure 9 with Figure 8, which only considers energy losses, we can see that the distribution of operations due to energy losses is slightly less uniform regarding the hourly distribution. However, this approach leads to a closer approximation to the optimal Cmax obtained in Figure 7.

## 8. Conclusions

This research work proposes a new strategy for simulating echolocation in order to address methods to solve real-time planning problems, taking advantage of the possibilities offered by new technologies in terms of real-time data acquisition and communication. This approach can serve as an alternative or as a complement to the methods used to solve this problem, instead of the approach of developing heuristic algorithms, which by definition can only provide approximate solutions to the problem.

An Industrial Sonar algorithm has been developed that can be used to optimise scheduling at different work centres while meeting the objectives of reducing the makespan, reducing the transport time of AMRs and reducing the energy costs and carbon footprint. Thus, this work presents a new methodology for real-time planning that combines different strategies to reduce production times and transport times, minimise energy costs and decrease the carbon footprint, and thus responds to a gap in the literature by integrating these variables and objectives in order to analyse and study production scheduling alternatives that incorporate operational and sustainable aspects into productive environments.

Furthermore, this new strategy can be used as a corrective strategy, to respond to difficulties that may arise during production, and also as a predictive and preventive simulation tool to design more flexible production processes capable of reacting quickly to any circumstance.

On the other hand, the Industrial Sonar algorithm proposes a flexible strategy that can be applied to any type of company or industry, adapting to the specificities of each business by using the most appropriate communication method and the most appropriate combination of algorithms at each work centre for that business.

As a potential future development, the real-time application of the Industrial Sonar algorithm between work centres can be compared and combined with other heuristic algorithms used to calculate the production schedule of all production centres. This would allow for conclusions to be drawn about the response times, the flexibility and the validity of the results in terms of time and energy losses.

As a future task, we propose applying this algorithm to various business processes to determine the most suitable communication methods for each process and the most effective combination of algorithms for each work centre.

As a suggestion for future research, the Industrial Sonar algorithm could be applied as a virtual tool for designing flexible production processes that can quickly adapt to any circumstance. This proposal would be specifically aimed at certain types of industry, but it could be extended to all.

As a future project, we propose linking the Industrial Sonar algorithm with MES/ERP production control programs. This will automate not only the programming in real time, but also its automatic application in production, including the management of the movement of AMRs between work centres.

**Author Contributions:** Conceptualisation, F.B. and M.-P.L.; methodology, F.B. and M.-P.L.; software, F.B.; validation, F.B. and M.-P.L.; formal analysis, F.B.; investigation, F.B. and M.-P.L.; data curation, F.B. and M.-P.L.; writing—original draft preparation, F.B.; writing—review and editing, F.B. and M.-P.L.; visualisation, F.B., M.-P.L., J.-A.R., P.M. and J.-C.S.; supervision, M.-P.L. and J.-A.R.; project administration, F.B., M.-P.L., J.-A.R., P.M. and J.-C.S. All authors have read and agreed to the published version of the manuscript.

**Funding:** This research received no external funding.

**Institutional Review Board Statement:** Not applicable.

**Informed Consent Statement:** Not applicable.

**Data Availability Statement:** The data presented in this study are available on request from the corresponding author. Data are unavailable due to privacy.

**Conflicts of Interest:** The authors declare that the research was conducted in the absence of any commercial or financial relationships that could be construed as a potential conflict of interest.

## Abbreviations

The following abbreviations are used in this manuscript:

| | |
|---|---|
| AMR | Autonomous Mobile Robot |
| AGV | Automatic Guided Vehicle |
| OEE | Overall Equipment Effectiveness |
| SPT | Shortest Process Time |

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
