# Peer review of "Real-Time Production Scheduling and Industrial Sonar and Their Application in Autonomous Mobile Robots"

_applsci, doi:10.3390/app14051890_

Round 1

Reviewer 1 Report

Comments and Suggestions for Authors

Thematically, the proposed manuscript is up-to-date. Solving the real-time optimal scheduling problem is one of the most challenging tasks in the field of distributed mechatronic systems. But I have the following more substantial remarks about the proposed article:

1. The working hypothesis in section 2 is not clearly defined

2. The title "Materials and methods" should be revised. The same applies to the text in this section, which reviews the state of research to date and the methods used. The emphasis should be on the comparison of methods in Table 1.

3. In section 5, emphasis should be placed on a more detailed description of the proposed industrial sonar algorithm.

4. Fig.1, which illustrates the architecture of industrial sonar, should be more understandable and unambiguous with its description.

5. The explanation of the figures is in the text, not in their title. This applies to all figures from 1 to 9.

6. When explaining the figures in the text, the corresponding figure is cited. All figures should be explained or discussed.

Comments on the Quality of English Language

Moderate editing of English language required

Reviewer 2 Report

Comments and Suggestions for Authors

The paper focuses on addressing the challenges and deviations in production scheduling caused by unexpected incidents. The research introduces the Industrial Sonar Algorithm, leveraging the biomimetic strategy of echolocation for real-time replanning in production centers. This algorithm aims to optimize the scheduling process by reducing makepan, energy costs, carbon footprint, and waiting and transport times for Autonomous Mobile Robots, using IoT, Cloud Computing, and Machine Learning technologies​​. Overall, the organization is well and the writing is clear. However, the following issues should be further considered and clearly clarified before the final publication:

1. In the Introduction, the paper emphasizes the importance of adapting to changes in real-time production scheduling. Could the authors elaborate on how the Industrial Sonar Algorithm specifically adapts to unpredictable changes in production environments, and how it compares to traditional methods in terms of flexibility and response times?​​

2. The methodology section describes using the Industrial Sonar concept for automatic communication between machines. Can the authors clarify how this method ensures robust and error-free communication, especially in complex industrial environments with numerous interacting machines?​​

3. In Sec. III, the paper discusses using sensors and business systems for real-time data collection. Could the authors provide more details on the types of sensors used, their integration with business systems, and how they contribute to the accuracy of the Industrial Sonar Algorithm?​​

4. In Sec. V, the Industrial Sonar method is proposed for interaction with work centers. How does this method ensure that the optimization of one work center does not adversely affect the overall production schedule, particularly in a multi-work center environment?​​

5. The Results and Discussion section presents the application of the Industrial Sonar method in a real environment. Could the authors provide a comparative analysis of the performance of this method against traditional scheduling methods, particularly in terms of efficiency and resource utilization?​​

6. In the Conclusions, future developments are discussed. Could the authors specify the potential limitations of the Industrial Sonar Algorithm and any anticipated challenges in its broader application across different industrial sectors?​​

7. 1.Since the authors considered a real-time Mobile Autonomous Robots based on IoT devices, it is suggested that the authors refer to some papers about the metric of data timeliness and mobile computing in your revised introduction part.

Comments on the Quality of English Language

It is important to proofread every part of the work to check for mistakes.

Reviewer 3 Report

Comments and Suggestions for Authors

1, The real-time production scheduling problem described in this paper is a very common problem and lacks noveltyï¼›

2, The method used in the paper is also a very simple one, and in fact, a lot of research has been conducted since the 1980s, some references are: (Duffie N,et al. Distributed Manufacturing Systems. J. Manuf.Syst, 1986,5(2): 137~ 139; Neil A. Duffie,Synthesis of Heterarchical manufacturing systems, Computers in Industry,Volume 14, Issues 1–3, May 1990, Pages 167-174; Smith, R. G. (1980). The contract net protocol: High level communication and control in a distributed problem solver. IEEE Transactions on Computer C, 29(12), 1104–1113.; N. Zbib, et al. Heterarchical production control in manufacturing systems using the potential fields concept, J Intell Manuf (2012) 23:1649–1670)

3, The work described in this article lacks deeper and more comprehensive exploration, and the concept and application of Industrial Sonar is relatively superficial.

4, The English expression in the paper is not good and there are many errors.

Comments on the Quality of English Language

Should be improved.

Reviewer 4 Report

Comments and Suggestions for Authors

The paper is very interesting but has many issues and therefore it is not suitable for publication at its current form. Authors need to critically revise the paper and address the following comments.

1) the structure of the paper is not clear. Authors need to revise the structure in order to be more clear. 

2) there is no literature review, authors need to search online for similar research works. The problem of dymanic rescheduling has been address in the literature, what is the difference from the proposed method. For example the paper below provides a solution for the dynamic rescheduling :

Psarommatis, F., Martiriggiano, G., Zheng, X. and Kiritsis, D., 2021. A generic methodology for calculating rescheduling time for multiple unexpected events in the era of zero defect manufacturing. Frontiers in Mechanical Engineering7, p.646507.

3) What is the novelty of the paper? this is not clear, as the paper is presetned now and without the literature review there is no novelty.

4) the proposed method is presented only in a theoretical basis, authors need to provide detaield desctiption of the algorithm , a flowchart, variables 

5)Figures 3,4,5,6 does not provide anyting to the reader as they are not readable.

6)Figures 7, 8 fonts are very small and difficult to read, please revise

7) What are the KPIs for the optimization of the schedules? why only energy?

8) what are the input data for the experiments? 

9) the results are mixed with the discussion, authors need to separate those sections.

10) how the proposed method is superior of those in the literature? no comparisons

Comments on the Quality of English Language

Minor english revisions are required

Reviewer 5 Report

Comments and Suggestions for Authors

The article is interesting but quite difficult to understand. I have a few specific comments for the authors to consider:

1. Is the presented concept original or does it result from the presented publication?

2. The conclusions are too limited and require further clarification What follows from the presented conclusions, can they have practical applications in some field of activity, e.g. industry, and if so, what specific ones? What potential outcomes might they apply to?

3. The bibliography is very poor, only 6 items are new, after 2020

4. Figures 2, 3, 4, 5 and 6 are completely illegible

Round 2

Reviewer 2 Report

Comments and Suggestions for Authors

1) It seems that the paper's format has some mistakes, such as the random highlight words and sentences. 2) The results from Figs. 7-9 need error analysis.

Comments on the Quality of English Language

The language of this work should be improved carefully. You should revise all mistakes: e.g, in the Introduction section: ' "...due to external factors intervene that'->'...due to external factors that'

Reviewer 4 Report

Comments and Suggestions for Authors

No further comments

Comments on the Quality of English Language

ok
